# Genomic and Metabolomic Insights into Secondary Metabolites of the Novel *Bacillus halotolerans* Hil4, an Endophyte with Promising Antagonistic Activity against Gray Mold and Plant Growth Promoting Potential

**DOI:** 10.3390/microorganisms9122508

**Published:** 2021-12-03

**Authors:** Eirini-Evangelia Thomloudi, Polina C. Tsalgatidou, Eirini Baira, Konstantinos Papadimitriou, Anastasia Venieraki, Panagiotis Katinakis

**Affiliations:** 1Laboratory of General and Agricultural Microbiology, Crop Science Department, Agricultural University of Athens, Iera Odos 75, 11855 Athens, Greece; e.e.thomloudi@gmail.com (E.-E.T.); polinatsal@gmail.com (P.C.T.); 2Department of Agriculture, University of the Peloponnese, 24100 Kalamata, Greece; 3Laboratory of Pesticides’ Toxicology, Department of Pesticides Control and Phytopharmacy, Benaki Phytopathological Institute, St. Delta 8, 14561 Athens, Greece; e.baira@bpi.gr; 4Department of Food Science and Technology, University of the Peloponnese, 24100 Kalamata, Greece; kostas.papadimitriou@gmail.com; 5Laboratory of Plant Pathology, Crop Science Department, Agricultural University of Athens, Iera Odos 75, 11855 Athens, Greece

**Keywords:** endophytes, plant growth promotion, biological control, postharvest biocontrol, biosynthetic gene clusters, *Bacillus halotolerans*, plant defense elicitors, grape, tomato, biopriming

## Abstract

The endophytic bacterial strain Hil4 was isolated from leaves of the medicinal plant *Hypericum hircinum*. It exhibited antifungal activity against *Botrytis cinerea* and a plethora of plant growth promoting traits in vitro. Whole genome sequencing revealed that it belongs to *Bacillus halotolerans* and possesses numerous secondary metabolite biosynthetic gene clusters and genes involved in plant growth promotion, colonization, and plant defense elicitation. The Mojavensin cluster was present in the genome, making this strain novel among plant-associated *B. halotolerans* strains. Extracts of secreted agar-diffusible compounds from single culture secretome extracts and dual cultures with *B. cinerea* were bioactive and had the same antifungal pattern on TLC plates after bioautography. UHPLC-HRMS analysis of the single culture secretome extract putatively annotated the consecutively produced antimicrobial substances and ISR elicitors. The isolate also proved efficient in minimizing the severity of gray mold post-harvest disease on table grape berries, as well as cherry tomatoes. Finally, it positively influenced the growth of *Arabidopsis thaliana* Col-0 and *Solanum lycopersicum* var. Chondrokatsari Messinias after seed biopriming in vitro. Overall, these results indicate that the *B. halotolerans* strain Hil4 is a promising novel plant growth promoting and biocontrol agent, and can be used in future research for the development of biostimulants and/or biological control agents.

## 1. Introduction

The adverse effects of the overuse of chemical pesticides and fertilizers on the environment and human health have created the need for alternative environmentally friendly strategies to minimize chemical inputs in agricultural practices, such as the use of beneficial plant-associated microorganisms called plant growth promoting microorganisms (PGPM) or plant probiotics [1]. They can enhance plant growth characteristics, mitigate abiotic plant stress, and/or protect plants from phytopathogenic microbes and pests, acting as biostimulants and/or biological control agents (BCAs) [2,3]. PGPM can solubilize nutrients, produce plant growth regulators and antagonize plant pathogens through the production of antimicrobial metabolites, the induction of plant defense, and space competition [2,3]. Plant growth promoting bacteria (PGPB) can colonize the rhizosphere, episphere, and/or endosphere [4]. Endophytes live inside plant tissues without causing disease and some can efficiently colonize the rhizosphere as well, while plant growth promoting endophytic bacteria are termed as PGPEB [4]. Medicinal plants seem to harbor endophytes with special characteristics and enhanced antagonistic activity against pathogens [5,6].

Plant-associated strains of the genus *Bacillus* have been studied and used commercially as biofertilizers and biological control agents, due to their ability to form thermostable and chemically resistant endospores, their vast secondary metabolite arsenal, their fast colonization ability, and their simple nutritional requirements [7,8]. However, the reported inconsistency of action has prompted an ongoing search for *Bacillus* plant probiotic strains and an in-depth investigation of strains’ genetic potential and biology. Endophytic bacteria are an interesting novel source of PGPB, since they achieve a more immediate effect on plants, they are protected from excess environmental stress and microbial antagonism, and they can inhibit latent pathogens, while the cultivable endophytes can also colonize the rhizosphere and rhizoplane, providing a broader site of action [4].

Endophytes have gained increased attention for use in agricultural applications due to their plant growth promoting and biological control abilities as well as in other fields, such as medicine and industry [9]. Endophytic bacteria can be very efficient biological control agents of pre- and post-harvest diseases [10], especially of pathogens that can remain cryptic (latent) in the endosphere such as *Botrytis cinerea*, which are detrimental to fruit crops worldwide [11]. Endophytic bacteria can combat pathogens by space occupation, production of lytic enzymes, and antifungal secondary metabolites [10,11], as well as through the activation of plant defense [12]. Studies regarding the biological control ability of bacteria in fruit or leaves are emerging in the literature for *B. velezensis*, *B. amyloliquefaciens* and *B. subtilis* strains [13,14,15,16,17]; however, studies for *B. haloterans* strains remain scarce [18,19,20].

This study aimed to (a) isolate endophytic bacteria residing in the leaves of an asymptomatic medicinal plant *Hypericum hircinum*, where plant bioactive compounds reside; (b) find the most antagonistic strain against numerous phytopathogens in vitro; (c) examine its plant growth promoting potential by biochemical and plant growth assays in vitro; (d) perform whole genome sequencing and genome mining of biosynthetic gene clusters or genes involved in plant growth promotion, biological control, and plant colonization; (e) investigate its biological control potential and mode of action against *B. cinerea* using in vitro assays and in vivo detached fruit assay; and (f) annotate the secreted diffusible metabolites.

## 2. Materials and Methods

### 2.1. Isolation of Endophytic Bacterial Strains

Asymptomatic medicinal plant *Hypericum hircinum* was collected from an open field belonging to the Agricultural University of Athens in Spata, Attiki, Greece. The surface sterilization of the excised leaves was performed according to [21]. Briefly, intact leaves were immersed in 70% ethanol for 1 min, followed by immersion in a solution containing 5% (*v*/*v*) aqueous solution of commercial bleach (5% sodium hypochlorite solution) and 0.1% Tween20 for 3 min. The tissues were immersed again in 70% ethanol for 30 s and then were rinsed thoroughly with sterilized double distilled water. The tissues were homogenized by making a paste using a mortar and pestle, which was plated on petri dishes containing nutrient agar (NA, Conda) amended with cycloheximide (100 μg/mL) and incubated at 28 °C for 2 weeks. Intact surface sterilized leaves and a quantity of the last wash were incubated as well, serving as controls for the success of the sterilization procedure. Bacteria were then subcultured to obtain pure cultures, originally aiming to include all different morphotypes and select one strain per colony morphology; however, only one colony type emerged. Strains were routinely maintained on NA at 4 °C and were stored long-term as a 40% glycerol stock at −80 °C and cultured in nutrient broth (NB, Conda) at 30 °C.

### 2.2. Assessment of Antifungal Activity In Vitro

All isolated strains were tested against the phytopathogenic fungus *B. cinerea* using dual culture assay in vitro on a NA medium (Conda), in order to select the most antagonistic strain. A mycelial disc (8 mm) was excised from an actively growing (7 days) fungal culture on PDA medium (Conda) and placed 3.5 cm across a 10 μL bacterial spot from an overnight NB culture. Each microorganism alone served as control. The plates were incubated at 25 °C for 15 days. There were 3 technical replicates and the experiment was performed 3 times.

The mode of action of endophytic bacterial strain Hil4 against *B. cinerea* was examined in vitro. The dual culture was repeated on NA and PDA media as described above and incubated for 9 days. The dual culture was also performed on a swarming NA medium, supplemented with 0.5% agar. The effect of secreted metabolites was studied by spotting filtered (0.22 μm) supernatant from a 48-h bacterial culture 2.5 cm away from a mycelial disc and incubated for 8 days. In all dual culture assays, each microorganism alone served as control; there were 3 replicates and 3 independent experiments. The effect of volatile compounds was investigated on the NA medium using I plates with a central partition. 10 μL of fungal spore suspension (10^5^ spores/mL) was spotted on one side of the petri dish, whereas 100 μL of bacterial culture was spotted on the other side. The plates were incubated at 25 °C for 5 days and checked daily. There were 10 replicates and the experiment was carried out 3 times. The production of biosurfactants was investigated using the rapid method of drop collapse with some modifications [22]. A 25μL drop of supernatant from a 48-h bacterial culture was spotted on parafilm and the degree of collapse was compared to a drop of water, using Evans Blue (0.01%) (Sigma-Aldrich^®^, Merk KGaA, Burlington, MA, USA) as visualization aid.

### 2.3. Evaluation of Plant Growth Promoting and Natural Fitting Traits In Vitro

All the isolated bacterial endophytes were screened for siderophore production by CAS agar assay [23], where 10 μL of an overnight culture was inoculated in the artificial well of the plate. The formation of an orange halo around the well, after incubation at 30 °C, indicates siderophore production. Production of indole related compounds was investigated using the Salkowski method [24]. Briefly, the supernatant from a 48-h bacterial culture in NB amended with 1% L-Tryptophan (Sigma-Aldrich^®^, Merk KGaA, Burlington, MA, USA), was mixed with Salkowski reagent in a 2:1 ratio, and was incubated in the dark for 30 min. A change in color to pink-orange-red indicated the production of indole-related compounds. The solubilization of precipitated phosphate was tested by inoculating 10 μL of overnight bacterial culture in an artificial well on the Pikovskaya medium containing tricalcium phosphate [25]. The formation of a clear halo around the well after incubation at 30 °C for 7 days indicated a positive result. Acetoin production was examined by the Voges–Proskauer test [26]. The appearance of a pink-red color in the liquid medium indicated acetoin production. Production of chitinase was tested on a NA medium amended with 1% colloidal chitin, prepared as described by [27]. Inoculation of 10 μL of bacterial overnight culture in an artificial well on the medium followed by the formation of a clear halo around the well after incubation at 30 °C for 2 days indicated chitin degradation. Cellulase production was examined by inoculating 10 μL of an overnight bacterial culture in an artificial well made on CYEA medium (0.5% casein, 0.25% yeast extract, 0.1% glucose, 1.8% agar) amended with 1% CMC (Carboxylmethyl cellulose) [24,28]. The formation of a clear halo around the well after incubation at 30 °C for 3 days indicated a positive result. To further confirm, the plate was flooded with Congo Red solution (1mg/mL), which was discarded after incubation for 15 min. Finally, the plate was flooded with NaCl solution (1M) to remove unbound Congo Red solution. Urease production was examined by spotting 10 μL of overnight bacterial culture in an artificial well on a Urea Base Christensen ISO 6579, ISO 19,250 (Conda, Madrid, Spain) medium [29]. The formation of a pink halo around the well indicated ureolytic properties. To test ammonia production, 10 μL of overnight culture was inoculated in a peptone water (4%) medium and incubated at 30 °C for 3 days [28]. Then, supernatant from this culture was mixed with Nessler’s reagent (Sigma-Aldrich^®^, Merk KGaA, Burlington, MA, USA) at a 10:1 ratio and incubated in the dark for 30 min. A change in color to brown-yellow indicated a positive result. Production of proteolytic enzymes was assayed on a CYEA medium (0.5% casein, 0.25% yeast extract, 0.1% glucose, 1.8% agar) amended with skim milk powder (7%) by inoculating 10 μL of an overnight bacterial culture in an artificial well and incubating at 30 °C for 2 days [30]. The formation of a clear halo around the well indicated production of proteases. The ability to perform swarming and swimming motility was investigated by spot inoculation of 10 μL from a bacterial overnight culture on a NA medium, amended with either 0.5% or 0.3% agar, and incubation at 30 °C for 1 day. Biofilm formation was determined using the crystal violet staining assay in 96-well PVC plates [31]. Briefly, 100 μL of a 1:100 dilution from a bacterial overnight culture was inoculated in each well (5 replicate wells) and incubated at 30 °C for 1 day without agitation. Then, the plate was rigorously washed 3 times with distilled water, then 200 μL of 0.1% (*w/v*) crystal violet were added in each well and the plate was incubated at room temperature for 30 min. The stain was removed by rigorous washing, 200 μL of a solution containing 20% (*v*/*v*) ethanol and 80% (*v*/*v*) acetone was added, and the plate was incubated at room temperature for 30 min to remove the biofilm formed by the stained cells from the walls of the well. The intensity of the purple color indicated the intensity of the biofilm formation. All assays were performed in triplicates in three independent experiments.

### 2.4. Identification of Endophytic Bacterial Strain Hil4 Based on 16S rDNA

Bacterial DNA was extracted from an overnight bacterial culture using a Nucleospin^®^ Microbial DNA (Macherey-Nagel GmbH & Co. KG, Düren, Germany) kit. Amplification of 16S rRNA gene fragments was performed by PCR using the primers F: 5′-AGAGTTTGATCCTGGCTCAG-3′, R: 5′-ACGGCTACCTTGTTACGACTT-3′ [32] and sequenced. BLASTn search was performed against the 16S database in GenBank of NCBI (https://blast.ncbi.nlm.nih.gov/Blast.cgi, accessed on 27 May 2021) to determine the genus and the closest phylogenetic group. The partial 16S rRNA gene sequence obtained was deposited in GenBank under the accession number MW672513.

### 2.5. Whole-Genome Sequencing

Genomic DNA was isolated from an overnight liquid culture of Bacillus strain Hil4 using the PureLink^®^ Genomic DNA Mini Kit (Thermo Fisher Scientific, Carlsbad, CA, USA) and was sequenced by SNPsaurus (Eugene, OR, USA) using an Illumina HiSeq 2000 platform. A library was generated using a Nextera XT DNA Library Prep Kit and the sequencing produced 2 × 150-bp paired-end reads that were trimmed with BBDuk and then assembled with SPAdes-3.12.0 using default parameters [33]. The final de novo genome of strain Hil4 (NCBI accession number: JAHHQB000000000) was assembled in 3 scaffolds with genome total size of 4,141,192 bp.

The strain’s phylogenetic identity was established by calculating the digital DNA:DNA hybridization (dDDH) using the genome-to-genome distance calculator website service (GGDC 2.1) under recommended formula [34] and the orthologous average nucleotide identity (OrthoANI) [35]. For ANI and dDDH analysis, species delineation threshold values were those suggested by default analysis (95–96% and 70%, respectively) [34,35]. Phylogenomic analysis of strain Hil4 was carried out by Type (strain) Genome Server (https://tygs.dsmz.de, accessed on 18 August 2021) and the tree was generated with FastME [36] from Genome BLAST Distance Phylogeny (GBDP) distances.

### 2.6. Functional Genome Analysis

Carbohydrate-active enzymes (CAZymes) were predicted in the core genomes using the server dbCAN2 and the combined tools HMMER (E-Value < 1e-15, coverage > 0.35), DIAMOND (E-Value < 1e-102) and Hotpep (Frequency > 2.6, Hits > 6) [37]. CAZyme domains predicted by at least two of the three algorithms (DIAMOND, HMMER, and Hotpep) were considered as true annotation for carbohydrate-active enzymes [37].

The online tool antiSMASH 6.0 [38] was used for secondary metabolites BGC prediction and annotation. The ClusterBlast and KnownClusterBlast modules integrated into antiSMASH 6.0 were also used for comparative gene cluster analysis based on the NCBI GenBank (https://www.ncbi.nlm.nih.gov/, accessed on 15 September 2021) and the ‘Minimum Information about a Biosynthetic Gene Cluster’ (MIBiG) [39] data standard, respectively.

### 2.7. Extraction of Secreted Diffusible Metabolites

The protocol followed for the extraction of Hill4 secreted agar diffusible metabolites (secretome) was based on [40,41] with modifications. Metabolites were extracted from Hil4 single cultures and Hil4-*Botrytis cinerea* dual cultures secretomes as described above. After 6 days of incubation at 30 °C, the confrontation zones from the dual cultures, as well as zones of the same weight and position from the single cultures, were excised with a sterile scalpel, cut into small pieces, and weighed. Then, they were placed in Erlenmayer flasks containing ethyl acetate (MilliporeSigma, Burlington, MA, USA) and 0.1% formic acid (MilliporeSigma, Burlington, MA, USA), vortexed, and sonicated in a water bath sonicator (Elmasonic S30H, Elma Schmidbauer GmbH, Singen, Germany) at room temperature for 30 min. The solution was then filtered through a Whatman filter and dried in a vacuum evaporator (Genevac HT-4, SP scientific, Ipswich, Suffolk, UK). Dried material was resuspended in HPLC grade methanol (MilliporeSigma, Burlington, MA, USA), filtered through Whatman^®^ Uniflo^®^ (MilliporeSigma, Burlington, MA, USA), and stored at −80 °C until further use.

### 2.8. Bioactivity of Extracted Diffusible Compounds against Botrytis cinerea Using Paper Disc Assay

To test the bioactivity of extracts from single bacterial culture (SC) and dual culture (DC) secretomes, 2 μL of *B. cinerea* spore suspension (10^5^ spores/mL) were inoculated on the center of NA plates. Extract impregnated discs (20 μL/disc) were placed 2 cm away from *B. cinerea* and the plates were incubated at 25 °C for 4 days. Methanol impregnated discs were placed in control plates. The formation of inhibition zones indicated the bioactivity of the extract.

### 2.9. Thin Layer Chromatography and Agar-Overlay Bioautography Method

Thin layer chromatography (TLC) of the single culture (SC) and dual culture (DC) secretome extracts was carried out based on [13]. Silica gel 60 F254 plates (20 × 20 cm; layer thickness, 0.20 mm; Merck) were used as the stationary phase and chloroform–methanol–water (65:25:4, *v*/*v*/*v*) as the mobile phase. For TLC-bioautography, developed chromatograms were overlaid with PDA medium containing 0.8% (*w*/*v*) agar and inoculated with *B. cinerea* spore suspension (10^5^ spores/mL). The TLC plate was incubated at 25 °C for 24 h and then sprayed with yellow tetrazolium salt dye solution (MTT, 2.5 mg/mL) (Sigma-Aldrich^®^, Merk KGaA, Burlington, MA, USA), which turns purple in the presence of living cells. After further incubation, the TLC plate was photographed and the Rf values of clear zones indicating antifungal activity were determined based on the formula: Rf = distance travelled by the solute/distance travelled by the solvent front.

### 2.10. Orbitrap High Resolution Mass Spectrometry (UHPLC-HRMS) Analysis

Chemical profiling of the single bacterial culture secretome extracts was investigated on an UHPLC-HRMS Orbitrap Q-Exactive platform (Thermo Scientific San Jose, CA, USA). A Hypersil Gold UPLC C18 (2.1 × 150 mm,1.9 μm) reversed phased column (Thermo Fisher Scientific, San Jose, CA, USA) was used for the analysis and maintained at 40 °C. Sample analysis was performed in positive (ESI+) and negative (ESI-) ion mode. Eluent A (ultrapure water with 0.1% formic acid) and B (acetonitrile) were used in a gradient mode of 30 min as follows: 0 to 21 min: 95% A, 5% B; 21 to 24 min: 5% A, 95% B; 24 to 30 min: 95% A, 5% B. The flow rate was 0.22 mL/min and data acquisition was performed on a mass range of 115–1500 m/z on profile mode. The conditions for the HRMS for both negative and positive modes were the following: capillary temperature, 350 °C; spray voltage, 2.7 kV; S-lense Rf level, 50 V; sheath gas flow, 40 arb. units; aux. gas flow, 5 arb. units; aux. gas heater temperature, 50 °C. The resolution for full scan analysis was set on 70,000, whereas for the data dependent acquisition mode, the resolution was 35,000, allowing for MS/MS fragmentation of the three most intense ions. Stepped normalized collision energy was set at 35, 60, and 100. After the acquisition, the data were processed using the Compound Discovered version 2.1 (Thermo Fisher Scientific, San Jose, CA, USA). For metabolite annotation, the online library mzCloud as well as the public chemical database PubChem (NCBI) were used, taking into consideration the isotopic and MS/MS fragmentation pattern and applying m/z tolerance of ±5 ppm.

### 2.11. Prevention of Gray Mold Disease of Grapes and Cherry Tomatoes and Colonization of Wounds

Table grape berries (*Vitis vinifera* var. soultana) and cherry tomatoes (acorn shape) (*Solanum lycopersicum* var. Lobello) were selected based on their homogeneity in size, maturity, and color, and the absence of visible injuries or infection. Fruit were thoroughly rinsed with tap water and distilled water and then surface sterilized by immersion in a 3% (*v*/*v*) aqueous solution of commercial bleach (5% sodium hypochlorite solution) for 20 min. Then, they were washed 5 times with sterilized distilled water and left to dry in a sterilized laminar flow cabinet. After drying, fruit were assigned randomly to each treatment and placed in the corresponding empty petri dish. An artificial wound was then inflicted in each fruit (3 mm × 3 mm for cherry tomatoes and 2 mm × 2 mm for grape berries).

For the grape berries assay, 5 μL of bacterial whole culture (vegetative cells, endospores, supernatant) containing 10^8^ CFU/mL (OD = 0.5) were inoculated in the wound of each berry and the petri dishes were transferred to sterilized transparent boxes containing wet filter paper to maintain moisture at high levels. After 36 h of incubation at 25 °C in the dark, 5 μL of *B. cinerea* aqueous spore suspension (10^5^ spores/mL) were inoculated in the same wound and the fruit were further incubated for 3 days. Controls of this assay were run in parallel with (a) only bacterial inoculant, (b) only fungal inoculant, and (c) only NB, applied at the appropriate time. The experiment consisted of 10 grape berries per treatment and it was repeated 3 times. The rating scale used to depict mycelial and rot coverage of each berry consisted of 4 classes: 0 = 0%, 1 = 0^+^ − 25%, 2 = 25^+^ − 50%, 3 = 50^+^ − 75%, 4 = 75^+^ − 100% [42].

For the cherry tomatoes assay, 10 μL of bacterial whole culture (vegetative cells, endospores, supernatant) containing 10^8^ CFU/mL (OD = 0.5) or diluted to 10^6^ CFU/mL were inoculated in the wound of each cherry tomato and the petri dishes were transferred to sterilized transparent boxes containing wet filter paper to maintain moisture at high levels. After 1.5 h of incubation at 25 °C in the dark, 10 μL of *B. cinerea* aqueous spore suspension (10^5^ spores/mL) were inoculated in the same wound and the tomatoes were further incubated for 3 days. Controls of this assay were run in parallel with (a) only bacterial inoculant, (b) only fungal inoculant, and (c) only NB, applied at the appropriate time. The experiment consisted of 12 cherry tomatoes per treatment and it was repeated 3 times. The rating scale used to depict mycelial coverage of each tomato consisted of 5 classes: 0 = 0%, 1 = 0^+^ − 10%, 2 = 10^+^ − 25%, 3 = 25^+^ − 50%, 4 = 50^+^ − 75%, 5 = 75^+^ − 100% [43].

Disease incidence (%) was calculated as the percentage of infected fruit, whereas the disease severity index was calculated based on the following formula [44,45]:(1)$$\mathrm{DSI}\;(\%)=\frac{\Sigma (di)}{DN}\;\times \;100$$
where *d* is the number of fruit in each class of the scale used, *i* is the number of the corresponding class, *N* is the total number of fruit, and *D* is the number of the highest class.

To examine the colonization of Hil4 in the wounds of grape berries and cherry tomatoes, the same experimental set up was followed, except the strain used was artificially resistant to rifampicin, by plating wild type strains on NA amended with gradient concentrations of rifampicin (25 μg/mL, 50 μg/mL, 100 μg/mL) and selecting one resistant colony. The protocol followed was based on [46]. After the 3-day incubation, tissue containing the wound of each fruit was excised using a sterilized scalpel, weighed, and ground in sterile NB using mortar and pestle. The homogenate was dilution-plated (serial decimal dilutions ranging from 10^0^ to 10^−9^) on NA plates amended with chloramphenicol (50 μg/mL), incubated at 30 °C for 1 day, and the colonies were counted to determine bacterial CFU per mg of tissue.

### 2.12. Effect on Growth and Root Morphology of Arabidopsis thaliana Col-0 In Vitro

Seeds of Arabidopsis thaliana Col-0 were kindly provided by Dr. Costas Delis from Department of Agriculture at the University of the Peloponnese. Surface sterilization of *A. thaliana* Col-0 seeds was performed by immersion in non-diluted commercial bleach (5% sodium hypochlorite solution) for 5 min and then rinsing 5 times with sterilized distilled water. Seeds were sown on half-strength Murashige and Skoog (½MS) medium, including vitamins (MS0222, Duchefa Biochemie, Haarlem, The Netherlands), containing 0.8% agar and 0.5% sucrose. After sowing, all plates were kept at 4 °C for 1 day and then were placed at an angle of 65° or horizontally (I plates) in a growth chamber (16-h light: 8-h dark photoperiod, 22 ± 1 °C, 50–60% relative humidity) until used for transplantation.

For the experiment of at distance inoculation, 4-day old seedlings, prepared as described above, were transplanted on ½ MS, including vitamins, containing 0.8% agar and 0.5% sucrose (6 seedlings/petri dish). Then, 5 μL of bacterial inoculant (10^8^ CFU/mL) were spotted at a 3.5 cm distance from each root tip and the plates were placed at an angle of 65° in the growth chamber for 12 more days. Control plants were treated with the same amount of sterilized distilled water. The plants were weighed and photographed in order to determine the length of the primary root, total lateral root number, and total root area. Stereoscopic images of the roots were used to determine root hair number and length in a 50 mm segment above the root tip. For the analysis of root hair length, the 20 longer root hairs from 6 plants were pooled (*n* = 120). There were 3 plates as replicates and the experiment was performed independently 3 times. Data from one representative experiment are shown.

For the experiment of on root tip inoculation, 6-day seedlings, prepared as described above, were transplanted on ½ MS, including vitamins, containing 0.8% agar and 0.5% sucrose (6 seedlings/petri dish). Then, 5 μL of bacterial inoculant (10^8^ CFU/mL) were spotted on each root tip and the plates were placed at an angle of 65° in the growth chamber for 10 more days. Control plants were treated with the same amount of sterilized distilled water. The plants were weighed and photographed in order to determine the length of the primary root.

To investigate the role of bacterial volatiles, surface sterilized seeds were sown on one half of an I plate containing ½MS including vitamins amended with 0.8% agar and 0.5% sucrose (4 seeds/petri dish) and were placed at 4 °C for 24 h. Then, 4 spots of 20 μL of bacterial inoculant (10^8^ CFU/mL) were inoculated on the other half. Control plants were treated with the same amount of sterilized distilled water. The plates were sealed with double parafilm, placed horizontally in a growth chamber (16-h light: 8-h dark photoperiod, 22 ± 1 °C, 50–60% relative humidity) and incubated for 25 days before fresh weight and leaf area were determined.

### 2.13. Effect on Germination and Growth on Solanum lycopersicum var. Chondrokatsari Messinias In Vitro by the Method of Biopriming

Seeds of *Solanum lycopersicum* var. Chondrokatsari Messinias were kindly provided by Dr. Costas Delis from Department of Agriculture at the University of the Peloponnese. The protocol followed is modified from [47,48]. All handling of the seeds was performed in a UV-sterilized laminar flow cabinet. Tomato seeds of variety Chondrokatsari Messinias were surface sterilized by immersion in 3% (*v*/*v*) aqueous solution of commercial bleach (5% sodium hypochlorite solution) for 6 min. Then, the seeds were washed thoroughly 6 times with sterilized double distilled water and left to dry. Dried seeds were immersed in a solution containing bacterial suspension (OD = 0.5) and 1% (*w*/*v*) Carboxyl Methyl Cellulose (CMC) (Sigma-Aldrich^®^, Merk KGaA, Burlington, MA, USA) and incubated at 25 °C for 1.5 h under agitation. Control seeds were immersed in a solution containing only CMC. After incubation, seeds were spread in empty petri dishes to dry and then placed on ½ MS, including vitamins, containing 0.8% agar (16 seeds/petri dish). The plates were incubated at 25 °C in the dark for 3 days and then placed with an open lid inside a sterilized transparent container and incubated in a growth chamber (16-h light: 8-h dark photoperiod, 25 ± 1 °C, 50–60% relative humidity) for 5 more days. Tomato seedlings were cleaned, weighed, and photographed to measure plant growth characteristics (seed germination, root length, shoot length, root area). There were 3 replicate plates and the experiment was performed 3 times.

The same procedure was followed in order to investigate the ability of the strain to colonize the roots of tomato after biopriming. Eight-day-old bioprimed seedlings were removed from the agar plate, cleaned with sterile filter paper, washed in sterile distilled water to remove loosely bound cells, and visualized under an Olympus BX40 microscope. The wash was dilution-plated in order to confirm the colonization of only the Hil4 strain.

### 2.14. Statistical Analysis

The software Fiji (https://imagej.net/software/fiji/, accessed on 20 August 2021) was used for measurements. Statistical analyses and plots were performed using GraphPad Prism version 9.0.0 for Windows (GraphPad Software, San Diego, CA, USA). The comparison of the bacterial treatment to the control was carried out by two-tailed independent samples Student’s *t* test (*p* < 0.05). Data of percentages were arcsine or logit transformed, while data on bacterial population were transformed to a logarithmic scale before being subjected to statistical analysis. Plots depict average values with standard deviation as error bars and asterisks indicate statistical differences.

## 3. Results

### 3.1. Isolation of Endophytic Bacteria from Hypericum hircinum

A total of 50 single bacterial colonies were isolated from leaves of the medicinal plant *H. hircinum*. These colonies were macroscopically and stereoscopically categorized based on colony morphology and only one category emerged. Among all the leaf isolates tested, Hil4 was the most effective strain against the fungal pathogen *Botrytis cinerea* in vitro, regarding the size of the inhibition zone, compared to the other endophytic isolates, so it was selected for further characterization. BLASTn analysis of the 16S rDNA sequence and comparison to type strains of the GenBank database of NCBI revealed this strain to be more closely related to *Bacillus halotolerans*, with a 98.29% identity to the *B. halotolerans* strain LMG 22477.

### 3.2. Endophytic Bacterial Strain Hil4 Possess Numerous Plant Growth Promoting and Environmental Fitting Traits

Strain Hil4 was tested for numerous plant growth promoting and environmental fitting traits in vitro (Figure 1). The isolate could produce indole-related compounds (Figure 1A), acetoin (Figure 1B), and ammonia (Figure 1C). It had also the ability to mobilize chelated iron (Figure 1D) and precipitated phosphate (Figure 1E) in vitro, as well as produce lytic enzymes such as chitinase (Figure 1F), protease (Figure 1G), cellulase (Figure 1I), but not urease (Figure 1H). Lastly, the endophytic isolate Hil4 proved to be an excellent swarmer (Figure 1J) with full plate coverage in 9 h, an excellent swimmer (Figure 1K) with full plate coverage in 15 h, and was capable of biofilm formation in polystyrene wells (Figure 1L).

### 3.3. Taxonomy and Genome Features of Endophytic Bacterial Strain Hil4

Strain Hil4 proved to be a promising plant growth promoting and biocontrol agent based on the in vitro screenings and was classified as *B. halotolerans* on 16S rDNA basis. Accurate taxonomic positioning, however, can only be achieved through whole genome sequencing and many species have been reclassified during the last years in the light of whole genome analysis [49,50]. Also, genome mining can reveal the potential of a strain for desirable attributes. Therefore, whole genome sequencing (WGS) was performed, followed by a phylogenome analysis. The comparison of strain Hil4 to other *Bacillus* strains at the genome level was carried out using the Type (strain) Genome Server (https://tygs.dsmz.de, accessed on 15 September 2021)) and revealed the close affiliation to the *B. halotolerans* species (Figure 2).

Further comparison was performed by alignment-based ANI and dDDH between strain Hil4 and other closely related *Bacillus* strains. The values obtained (Table 1) were above the threshold for species delineation in both methods regarding *B. halotolerans*, ranging between 82.10 and 93% for dDDH and 98.05 and 99.14% for ANI analysis. As far as *B. mojavensis* strains are concerned, OrthoANIu values were equal to the cut-off values, while dDDH values were below the cut-off values. ANI values 95–96% and dDDH values are considered as threshold values for species separation [34,35]. Since both ANI and dDDH methods, as well as the phylogenome analysis were in agreement, endophytic bacterial strain Hil4 was classified as a *B. halotolerans* species.

### 3.4. Bacterial Strain Hil4 Possesses Secondary Metabolite Biosynthetic Gene Clusters

The genome of bacterial strain Hil4 was further analyzed using antiSMASH in order to predict and locate biosynthetic gene clusters (BGCs) of secondary metabolites. Based on the antiSMASH analysis, bacterial stain Hil4 harbors 11 individual BGCs (Table 2), across 10 biosynthetic regions, with region 3 comprising two BGC classes, covering approximately 15% of the whole genome (Table 2). These include BGCs producing known compounds, as well as compounds novel or not registered in the MIBiG database. Two BGCs of terpenes and one BGC containing a Type III polyketide synthetase had no match in the known compounds of the database, but were common in other strains of *B. halotolerans*, *Bacillus subtilis*, and *Bacillus valismortis* based on antiSMASH ClusterBlast analysis, with 100% gene similarity (Figure 3). A fourth BGC had 28% similarity with a known BGC encoding for the synthetases of the polyketide aurantinin. The antiSMASH ClusterBlast analysis revealed that the putative aurantinin cluster of strain Hil4 shared high gene similarity (69%) with a newly identified FAS-PKS gene cluster found in *B. velezensis* SQR9, while nucleotide alignment of the biosynthetic genes yielded 97.7% identity (Figure 3). In this regard, the Hil4 aurantinin BGC will be referred to as the putative FAS-PKS gene cluster (Table 2, Figure 3). The remaining known clusters presented 100% homology with the deposited BGCs and similar gene organization with other *B. halotolerans* strains (Table 2, Figure 3). They represent the biosynthetic genes encoding for synthetases of surfactin, fengycin and bacillibactin (non-ribosomal peptides), bacillaene and mojavensin (hybrid; polyketide and non-ribosomal peptide), bacilysin (non-ribosomally synthesized antibiotic), as well as the bacteriocin subtilosin (lanthipeptide). It should be noted that mojavensin BGC is adjacent to fengycin BGC and both are located in the genomic region 3. Using the antiSMASH, the mojavensin cluster was presented as an iturin-type cluster, showing 100% gene homology to the mycosubtilin BGC, 60% gene homology to the bacillomycin D BGC, and 44% gene homology to the iturin BGC. However, the predicted aminoacid sequence (Figure 3) indicated that the BGC encodes for the synthetases of mojavensin, due to the presence of the module C Asn_6_ and Asn_7_ [52].

An interesting discovery while searching the mojavensin cluster in other *B. halotolerans* strains through antiSMASH and BLAST analysis was that not all strains harbor this BGC. For example, the mojavensin gene cluster is present in *B. halotolerans* FJAT-2398, but absent from *B. halotolerans* KKD1. As shown in the microsynteny maps of fengycin/mojavensin BGC genome regions (Figure 4), genes of the flanking regions 2 and 3 in the genome of FJAT-2398 were also present in the genome of KKD1, as well as in the genome of strain Hil4. However, in the KKD1 genome, flanking region 3 is located right after flanking region 2, while in the two other *B. halotolerans* strains, mojavensin BGC interrupts this continuity, suggesting that mojavensin BGC may have been inserted through horizontal transfer events (Figure 4). Another interesting finding through BLASTp analysis concerning Hil4 mojavensin BGC evolution was that Bacillus cabrialesii strain BSIIRRCK3 harbors mojavensin non-ribosomal protein synthetases (NRPS) highly similar (>98% identity at protein level) to Hil4 mojavensin NRPS, although the two species are phylogenetically distinct (Figure 2, Table 1). Comparison of Hil4 mojavensin BGC genomic region to the one from *B. cabrialesii* strain BSIIRRCK3 revealed identical gene organization in the flanking regions as well as the two biosynthetic gene clusters (Figure 4). Nucleotide alignment of the genomic regions of the two species showed 81% nucleotide identity of the fengycin core biosynthetic gene cluster and 99% nucleotide identity of the mojavensin core biosynthetic gene cluster, while flaking regions 1, 2, and 3 shared 86.57%, 85.60%, and 87.99% nucleotide identity, respectively. The possible insertion site of mojavensin BGC is depicted in Figure 4 and appear to be conserved in both mojavensin-harboring *B. halotolerans* and *B. cabrialensis* strains. For all strains analyzed, all genes of the flanking regions were conserved with the exception of three genes present or absent in each strain, shown as blue asterisks in the example of Figure 4.

### 3.5. Bacterial Strain Hil4 Possesses Genes Involved in Biological Control, Plant Growth, and Colonization

Besides the BGCs that might be involved in biosynthesis of antifungal compounds and/or plant defense elicitors, genome mining also revealed other genes involved in biocontrol, plant growth, and colonization (Appendix A). Genes responsible for plant growth regulators such as auxin, nitric oxide and spermidine, or biocontrol such as acetoin and 2,3 butanediol were found along with enzymes aiding nutrient solubilization such as phytase and phosphatase. Also, the analysis revealed genes involved in chemotactic motility, colonization and adherence such as flagella biosynthesis, swarming motility, and biofilm formation.

In addition, a total of 126 genes encoding CAZy (carbohydrate-active) enzymes were identified in the genome of *B. halotolerans* Hil4. A fraction of 51 belonged to glycoside hydrolases (GHs), 41 belonged to glycosyltransferases (GTs), 7 belonged to polysaccharide lyases (PLs), 12 belonged to carbohydrate esterases (CEs) and 1 belonged to the category auxiliary activities (AAs), while 14 carbohydrate-binding modules (CBMs) were predicted. A large fraction (24.60%) of the CAZyme proteins possess an amino-terminal signal peptide for mediating export of proteins across the cytoplasmic membrane, indicating that they are secreted enzymes. The CAZy enzymes included enzymes targeting fungal cell walls such as chitinase (GH18), chitosanase (GH46), glucanase (GH51, GH16), plant cell walls such as β-glucosidase (GH1), pectate lyase (PL9, PL3), and pectin acetylesterase (CE12), as well as starch such as a-amylase (GH13).

### 3.6. Bacterial Strain Hil4 Antagonizes Botrytis cinerea through Secretion of Diffusible Metabolites In Vitro

Strain Hil4 showed antagonistic activity against *B. cinerea* both on NA (Figure 5A) and PDA (Figure 5A) media, with the formation of an inhibition zone after 9 days of incubation. 

When co-cultured with *B. cinerea* on the PDA medium, a white precipitate was formed around the bacterial colony, that was not present on the NA medium or when the antagonist and pathogen were cultured alone in either growth medium. The formation of the inhibition zone on both media, as well as the white precipitant led to the hypothesis that secreted diffusible metabolites were the main antagonistic mode of action of this strain. The culture supernatant showed biosurfactant properties in the drop collapse assay (Figure 5B), indicating the presence of antifungal lipopeptides. When cell free culture supernatant was tested in a dual culture assay, it had an inhibitory effect by forming an inhibition zone for 4 days; then the phytopathogen slowly overgrew it (Figure 5C). Based on these data, the activity of the bacterial strain Hil4 against *B. cinerea* in vitro seems to be the result of antifungal metabolite secretion.

### 3.7. Ethyl Acetate Extracts of Secreted Diffusible Metabolites from Solid Culture Are Bioactive Against Botrytis cinerea in Dual Culture and TLC-Agar Overlay Bioautography Analysis

Considering the antagonistic nature of bacterial strain Hil4 against *B. cinerea* in vitro, secreted diffusible metabolites were isolated in order to be identified. Ethyl acetate extracts from single bacterial culture and dual bacterial-fungal culture secretomes were prepared in order to investigate whether agar diffusible metabolites were produced constitutively by the bacterial strain or upon fungal recognition.

Both single culture and dual culture extracts exhibited antagonistic activity against *B. cinerea* in vitro (Figure 6A). When the extracts were analyzed using thin layer chromatography, it was observed that both extracts had the same pattern regarding the antifungal zones formed after the agar overlay bioautography, with similar retention factors (Rf) (Figure 6B). Specifically, two zones appeared, with Rf1 = 0.367 and Rf2 = 0.409 for the extract from bacterial single culture secretomes (SC) and Rf1 = 0.359 and Rf2 = 0.401 for the extract from fungal-bacterial dual culture (DC).

### 3.8. Putatively Annotation of Antimicrobial Compounds and ISR Elicitors in Ethyl Acetate Extracts of Secreted Diffusible Metabolites from Solid Hil4 Culture through UHPLC-HRMS Analysis

Secreted diffusible metabolites present in the ethyl acetate extract of single bacterial culture secretome were annotated using UHPLC-HRMS chemical analysis, which revealed several antimicrobial compounds and plant defense elicitors (Table 3).

The annotation of the compounds was based on online databases (mzCloud, NCBI etc.) and already published data, taking into consideration the isotope distribution and the accurate mass (±5 ppm). The chromatograms are depicted in Figure 7. 

The accurate masses obtained from ions of 1006.6448 *m/z* and 1034.6760 *m/z* in the negative ion mode were putatively annotated as two isoforms of the cyclic lipopeptide Surfactin A, with molecular types C_51_H_89_N_7_O_13_ (C13) and C_53_H_93_N_7_O_13_ (C15), respectively. The 1082.5618 *m/z* was annotated to the compound Mojavensin A with molecular formula C_50_H_77_N_13_O_14_. This analysis also detected a peak at 1461.7890 *m/z* putatively corresponding to the molecular formula C_72_H_110_N_12_O_20_, indicating the presence of the cyclic lipopeptide Fengycin A C16. In addition, the compound with 579.3445 *m/z* was tentatively annotated as Bacillaene A1 with a molecular formula of C_34_H_48_N_2_O_6_. Another compound was detected at a 200.0923 *m/z* with putatively molecular type C_9_H_15_NO_4_, corresponding to the compound L-dihydroanticapsin. A compound with 881.2488 *m/z* and possible molecular type C_39_H_42_N_6_O_18_ was attributed to the siderophore Bacillibactin. Lastly, the compound with 187.0968 *m/z* was annotated as azelaic acid with the molecular formula of C_9_H_16_O_4_.

### 3.9. Bacterial Strain Hil4 Lowers Disease Severity of Gray Mold on Grape Berries in Preventive Application

Following the discovery that bacterial strain Hil4 produces antimicrobial compounds and plant defense elicitors, it was tested whether it could suppress the gray mold disease caused by the phytopathogen *B. cinerea* on detached grape berries when applied as a preventive treatment. Bacterial culture suspension (10^8^ CFU/mL) was inoculated in an artificial inflicted wound and incubated for 36 h before applying the phytopathogen, since incubation of lesser time resulted in no disease inhibition. The disease suppression was evident in grape berries (Figure 8A) and the strain efficiently colonized the grape berries either when applied singly or when applied before *B. cinerea* infection (Figure 8B). Although Hil4 did not lower disease incidence (Figure 8C), it achieved a statistically significant decrease in disease severity of approximately 30% (Figure 8D).

### 3.10. Endophytic Strain Hil4 Protects Cherry Tomatoes from Gray Mold in Preventive Application

The effectiveness of the bacterial strain Hil4 against gray mold was also investigated in detached cherry tomatoes. Bacterial suspension was inoculated in an artificially inflicted wound and after 1.5 h of incubation, the pathogen was inoculated in the same wound. When bacterial suspension of 10^8^ CFU/mL was used as inoculant, a rapid bacterial growth and the formation of protective biofilm was observed in the wound, resulting in a significant reduction of disease incidence (approximately 70%) (Figure 9A). Subsequently, a bacterial concentration of 10^6^ CFU/mL was used as inoculant, where overgrowth was not observed while the strain efficiently colonized the wound (Figure 9B). Although disease incidence was not decreased at this bacterial concentration, disease severity was decreased by 25.5% (Figure 9B).

### 3.11. Bacterial Strain Hil4 Influences Growth and Root Architecture of Arabidopsis thaliana Col-0 with Different Inoculation Methods In Vitro

The isolate Hil4 was also examined for its effect on growth of *Arabidopsis thaliana* Col-0 in vitro. To investigate whether diffusible and/or volatile compounds affect plant growth, a bacterial suspension was inoculated at distance from the root tip, on the root tip, and in a different compartment of the petri dish.

As shown in Figure 10, the inoculation of isolate Hil4 at distance from the root tip had a statistically significant impact on measured plant growth parameters compared to the untreated control (Figure 10A). Treated plants had higher fresh weight (Figure 10B), shorter primary root (Figure 10C), increased root area (Figure 10D), and lateral roots (Figure 10E). The most prominent morphological alteration in the root system architecture seems to be the stimulation of lateral root formation, resulting in an increase in total root area despite the shortening of the primary root. Also, root hairs located on the root tips increased in number (Figure 10F) as well as length (Figure 10G). When inoculated on the root tip, strain Hil4 provoked evident differences in the phenotype of treated plants compared to the untreated control (Figure 10H), with a statistically significant increase in fresh weight (Figure 10I) as well as a statistically significant decrease in primary root length (Figure 10J) compared to the control plants.

Another experiment on *A. thaliana* Col-0 consisted of examining the impact of bacterial volatiles on the plant growth. Therefore, I plates with central partitions were implemented, in which plants grow in one compartment and inoculant grows in the other, eliminating contact or diffusible compounds (Figure 11A). A small increase in fresh weight was observed in treated plants compared to the control (Figure 11B), whereas a large increase in leaf area was observed in treated plants compared to the control (Figure 11C).

### 3.12. Bacterial Strain Hil4 Promotes Growth and Early Germination of Solanum lycopersicum var. Chondrokatsari Messinias after Seed Biopriming

After validating that the endophytic bacterial strain had an effect on growth and root morphology of the model plant *A. thaliana* Col-0, its effect was examined on tomato under in vitro conditions. Seeds of *Solanum lycopersicum* var. Chondrokatsari Messinias were selected, which is a native Greek variety [61], while the bacterial inoculation was carried out by the method of seed biopriming using CMC (1%) as adhesive.

Biopriming of seeds with the bacterial strain Hil4 had an evident effect on plant developmental characteristics (Figure 12A). It was also established that this strain could colonize the root after seedling emergence, according to dilution plating as well as microscopic observation, which showed that bacterial cells were either dispersed or aggregated on the root surface (Figure 12B). A small increase in germination percentage was observed compared to the control treatment at 3 days of incubation, but not at 8 days, suggesting that earlier germination was achieved instead of an overall enhancement of germination. Bacterial strain Hil4 enhanced all plant growth parameters examined—fresh weight, shoot length, root length, and root area—in a statistically significant way compared to control plants.

## 4. Discussion

*Bacillus* species have been widely used as plant inoculants for sustainable agriculture due to their ability to promote plant growth, act as biocontrol agents, and possess traits associated with environmental fitness such as swarming motility and endospore formation [62]. In addition, bacteria that can adapt to an endophytic lifestyle are considered ideal for the suppression of systemic phytopathogens such as *B. cinerea* [11]. In this study, strain Hil4 proved to be a novel plant-associated *B. halotolerans* strain regarding its genetic potential that exerted plant growth promoting and biocontrol effects.

Culturable endophytic bacteria were isolated from leaves of the medicinal plant *Hypericum hircinum* and only one colony type emerged. Bacterial diversity in the endosphere of leaves is usually reduced compared to the roots and is also dependent on numerous parameters, such as the plant species and age [63]. Among the individual colonies tested, isolate Hil4 exhibited the largest inhibition zone against *B. cinerea* in vitro and was selected for further characterization. It was also found positive in numerous biochemical tests regarding plant growth promoting and environmental fitting traits; it could produce indole-related compounds such as acetoin and ammonia, mobilize chelated iron, solubilize precipitated phosphate, and produce an array of lytic enzymes, such as chitinase, cellulose, and protease. All these traits that are possibly active in the rhizosphere and endosphere can alter plant growth, make nutrients bioavailable, and protect from phytopathogens [2]. The production of indole-related compounds is considered an indication of indoloacetic acid production, which is the plant hormone auxin [29,64]. This stain is also capable of fast swarming and swimming motility, as well as biofilm formation, environmental fitting traits that help with niche occupancy, plant colonization, and adherence, as well as outperforming other microbes in the vicinity [65,66].

To taxonomically characterize this strain, 16S rDNA sequencing was performed that classified it as *B. halotolerans*. The recent revolution of whole genome sequencing (WGS) has resulted in reclassifications of many strains across Bacillus species, showcasing it as the most reliable method of taxonomy. For example, *Brevibacterium halotolerans*, *Bacillus axarquiensis*, and *Bacillus malacitensis* were reassigned as *B. halotolerans*, as well as the *B. mojavensis* strains [49]. In addition, genome mining provides information for key bacterial characteristics, revealing the genetic potential of the isolate, so WGS was performed on the bacterial strain Hil4. Phylogenome analysis as well as obtained ANI and dDDH values classified endophytic bacterial strain Hil4 as a *B. halotolerans* strain. Genome mining of putative BGCs of secondary metabolites revealed that the strain has the potential to produce a large arsenal of antimicrobial compounds and ISR elicitors. The antiSMASH analysis predicted 11 putative BGCs for secondary metabolites: 4 unknown as well as 7 known. The Hil4 putative FAS-PKS BGC showed high gene similarity to a FAS-PKS cluster from *B. velezensis* SQR9, which is responsible for the biosynthesis of bacillunoic acids, novel fatty acids with strong antibacterial ability [67]. Bacillaene is an inhibitor of prokaryotic protein synthesis, indicating antagonism with other prokaryotes in the vicinity, and seems to protect B. subtilis from predation [68,69]. Subtilosin belongs to class I bacteriocins, called lantibiotics, and displays antimicrobial activity against gram positive and negative bacteria; strains producing it are being researched as animal or human probiotics [70,71]. Bacilysin is a dipeptide antibiotic that shows high antibacterial activity against bacteria such as *Erwinia amylovora* and *Xanthomonas* spp. that cause important plant diseases [72,73], as well as against *Candida albicans*. Bacillibactin is a siderophore synthesized by non-ribosomal peptide synthatases and is a high affinity Fe3+ chelator able to contribute to plant nutrition and make iron less available to pathogenic fungi [74] and act in direct antibiosis [75]. Bacillibactin has been reported to inhibit growth and plant invasion of *Phytopthora capsici* and *Fusarium oxysporum* and lower disease severity [76,77]. The ability of this strain to produce siderophores was confirmed by a positive result in the CAS-agar assay in vitro. Surfactins belong to cyclic lipopeptides, are strong biosurfactants, have proved essential for biofilm formation, swarming motility and root colonization and exhibit haemolytic, antibacterial, and antiviral activities [78]. Fengycins are cyclic lipopeptides with strong antifungal activity against filamentous fungi [78]. The fengycin BGC was adjacent to an iturinic-type BGC in the genome of strain Hil4, with 100% gene similarity to mycosubtilin. However, it proved to be mojavensin BGC, not registered in the MIBiG database, based on the predicted amino acid sequence of the final product (Asp6, Asp7) [52]. Mojavensin belongs to the iturin family of cyclic lipopeptides and was recently isolated from the fermentation broth of the marine *Bacillus mojavensis* B0621A [55,79]. It seems to exhibit low antifungal activity (MIC = 2 mg/mL), but its exact functional role is not elucidated yet. Based on antiSMASH ClusterBlast analysis, some *B. halotolerans* strains harbored the BGC, while others did not. However, the gene organization of the flanking regions and the core biosynthetic genes was almost identical among strains, suggesting the ability of the mojavensin cluster to enter the genome in a specific genomic location.

Previous studies have shown that mojavensin BGC and the putative FAS-PKS BGC seems to be absent from the published genomes of plant-associated *B. halotolerans* [18,80,81,82], thus underlining the novelty of the endophytic strain Hil4. Our studies provided the first example that clustered genes such as mojavensin BGC are jointly transferred in *B. halotolerans* species and inserted in specific regions that may represent hot spots for BGC acquisition. Furthermore, our studies provided, to the best of our knowledge, the first example of a possible BGC exchange between *B. halotolerans* and *B. cabrialesii*. The core biosynthetic genes of mojavensin in the genome of strain Hil4 also shared an extensive sequence identity in *B. cabrialesii* BSIIRRCK3. However, *B. cabrialesii* BSIIRRCK3 is distantly related in terms of phylogeny to *B. halotolerans* Hil4, suggesting that the mojavensin BGC might have been horizontally exchanged between these bacteria. The possible BGC exchange of the FAS-PKS cluster of bacillunoic acids between strains of *B. velezensis* has been examined [67], with the genomic region being reported as a ‘hot spot site for acquiring horizontally transferred genes’, since all examined *B. velezensis* strains harbored genomic islands that were similar or different compared to the one from strain SQR9, while *B. subtilis* ATCC 19,217 harbored an identical genomic region. There is substantial evidence that BGCs are horizontally exchanged among bacteria [83,84,85], although the frequency at which this occurs and the mechanisms driving these events remain poorly understood.

Nonetheless, horizontal gene transfer is widely recognized as a driving force in bacterial evolution and the acquisition of a gene cluster encoding the production of a specialized metabolite could be advantageous to a bacterial strain in terms of more effective multiplication, escape from predation, or accession to more resources, compared to strains without this BGC [86].

Further genome mining revealed high enrichment of PGP and biocontrol potential abilities as genes involved in motility, biofilm formation, plant growth promotion, and plant defense were identified, whose products, such as indole-related compounds (possibly auxin) and acetoin, were detected through in vitro biochemical assays. Furthermore, CAZymes analysis revealed the potential of lytic enzymes production to aid in endophytic colonization and fungal inhibition, such as chitinase that was proved to be secreted in vitro. The secretion of chitinase and chitosanase could contribute in the enhancement of the direct control of phytopathogenic fungi [87] or indirect control, where the chitosan fragments released through the action of chitinolytic enzymes elicit plant defense reactions both locally and systemically [88].

Endophytic bacterial strain Hil4 strongly inhibited *B. cinerea* in vitro, on both NA and PDA media. The culture supernant of Hil4 was positive in the drop collapse assay, indicating the presence of biosurfactant cyclic lipopeptides, and was active against the phytopathogen, although for fewer days than the bacterial colony. During dual culture on PDA medium, a white precipitate was observed around the bacterial colony. Formation of white precipitates surrounding bacterial colonies in the area of the inhibition zone has previously been observed by [89] when testing endospores of *B. velezensis* FZB42 against *B. cinerea*. The authors suggested that it could originate from co-precipitation of the secreted lipopeptides once they reach a certain concentration, resulting in lower amounts of soluble iturin and fengycin diffusing in the medium and involved in pathogen arrest [89]. Many necrotrophic pathogens, including grey mold, produce oxalic acid, which is a strong chelator of cations, oxidizes organic compounds, prevents defense signaling pathways in the plant, and reduces pH, thereby facilitating the precipitation of secreted cyclic lipopepides [89] and the action of pectolytic enzymes [90]. The formation of white precipitate has also been reported in the study of [91] where vegetative cells and endospores of *B. amyloliquefaciens* BUZ-14 were tested against *B. cinerea* and other phytopathogens on PDA medium. In our experiments, the precipitate was present when using PDA as substrate, but not NA, a fact that emphasizes this hypothesis, as PDA supports better fungal growth and perhaps the greater production of oxalates. Furthermore, when bioactive supernatant was tested on PDA medium, there no precipitant formed and a smaller inhibition zone was observed, maybe due to the smaller amount of compounds present compared to the active production of a growing bacterial colony. The oxalate decarboxylase gene oxdC is present in the genome of the bacterial strain Hil4; however, white precipitate due to oxalic acid managed to form, suggesting that the gene is either not expressed or inefficient in inactivation of the produced oxalic acid.

In order to investigate whether secondary metabolite BGCs found in the genome of strain Hil4 are active and expressed constitutively, ethyl extracts containing secreted agar-diffusible metabolites from solid cultures of Hil4 singly grown or dual cultured with *B. cinerea* were analyzed. Ethyl acetate extracts have been reported to contain more metabolites than other type extracts [92]. It was found that both extracts had the same pattern after TLC development and bioautography, with Rf values corresponding to iturinic lipopeptides [93,94]. TLC is a poor method of identification, since spot development and consequently bioactivity depend on conditions such as humidity and temperature [95], and there are inconsistencies regarding R_f_ values for iturinic lipopeptides [13,92,93,94]. Therefore, the extract of the agar-diffused metabolites from single solid culture secretome of Hil4 was further analyzed with UHPLC-HRMS. The possible annotation of the compounds was based on previously published data. Numerous antimicrobial metabolites and ISR elicitors were annotated, indicating that this strain has the ability to produce them constitutively, a very important aspect in biological control [96]. The metabolites putatively annotated were surfactin with two homologues (C13, C15), mojavensin A, fengycin (C16), bacillaene, and the siderophore bacillibactin. L-dihydroanticapsin, a precursor of bacilysin and the active antifungal compound against *Candida albicans* [97], was also tentatively annotated in the extract. The compound subtilosin was not detected, while a novel compound was present in the extract: azelaic acid. Azelaic acid is a dicarboxylic acid that is found in cereals, rye, and barley [98], and is suggested to move systemically, contributing to plant defense [99]. It was also recently found that supernatant from *Pseudomonas syringae* contains azelaic acid, suggesting a role in plant-bacteria communication [60]. The first report of azelaic acid production from *Bacillus* species was in a study of *B. velezensis* Bvel1 that also examined ethyl acetate extracts containing secreted agar-diffusible metabolites from singly grown bacterial solid culture [15], as in the present study.

Since *B. halotolerans* Hil4 demonstrated high antifungal activity and constitutively produced antimicrobial metabolites and ISR elicitors in vitro, it was investigated whether it could mitigate disease symptoms of gray mold in cherry tomatoes and grape berries. Previous studies have shown that the application of *Bacillus* strains is a promising method for decreasing the decay of harvested fruit due to their spore-forming abilities and secondary metabolite reservoir [14,16,100]. Many *Bacillus* strains are also considered as safe to be used as animal probiotics or in food preservation [101,102,103]. However, *B. halotolerans* species have not been studied to such an extent regarding post-harvest diseases in fruit [18,19] and as far as we are aware, there is not a study with the species involving gray mold of tomato fruit or grapes.

The application of bacterial cell suspension could be more advantageous in controlling plant pathogens than applying cell-free supernatants or purified compounds [14,104], although some studies have reported otherwise [105]. The superiority of cell suspension treatment, with the prerequisite of good colonization, could be due to the continuous production of metabolites, the space competition of bacterial cells, the defense responses cells and secondary metabolites could provoke, and the co-production of a large arsenal of metabolites, as well as the synergistic effects between these factors [78,96].

It has also been suggested that preventive treatment is more efficient than curative treatment (either in the field or post-harvest) [16,91,105]. Therefore, the effect of preventive bacterial cell application was examined in the current study. Following Hil4 culture suspension application to grapes with 10^8^ CFU/mL, disease incidence was not affected compared to the control, but disease severity was decreased by approximately 30%. The same trend was observed during the experiment with cherry tomatoes when the strain was inoculated at a concentration of 10^6^ CFU/mL, while the disease incidence was decreased when the strain was inoculated at a concentration of 10^8^ CFU/mL. This is indicative of this strain’s ability to act as a biological control against diseases in fruit in artificial conditions with a large fungal inoculum, suggesting future experiments with unwounded fruit in field conditions for optimizing the bacterial concentration and time of application. Nevertheless, this is the first study reporting a decrease in the disease severity of gray mold in cherry tomatoes and grapes due to a *B. halotolerans* strain application. Previous studies have demonstrated that *B. halotolerans* KLBC XJ-5 treatment did not lower the disease incidence in strawberry at 4 days of incubation compared to the control treatment, but significantly decreased the lesion diameter (10.9 mm) compared to the control (14.5 mm) [19]. Additionally, in the study [18], *B. halotolerans* strain BFOA4 was reported to lower the disease severity of tomato fruit caused by *Fusarium oxysporum* f.sp. *radicis-lycopersici.*

Numerous studies have investigated the mode of action of biocontrol agents on post-harvest disease suppression. Except from suggested direct antibiosis [56,105,106], suppression is also attributed to indirect activity through activation of plant defense pathways [16,19,107]. *Bacillus* lipopeptides such as iturins and fengycins are known to possess strong antifungal activity by forming ion pore channels in the membranes [108] and can also act as plant defense elicitors in detached fruit [109,110]. In the study [111], fengycin was proved to be an important factor in the biocontrol of apple ring rot by *B. subtilis* 9407. In addition, other compounds, such as antibacterial metabolites, can act as plant ISR elicitors [99,112,113], as well as the compounds released from the action of lytic enzymes on the plant cell wall [19] or endophytic bacterial cells themselves [12]. Furthermore, plant growth regulators can be plant defense determinants, such as acetoin [113,114]. The genomic potential of bacterial strain Hil4 to produce several metabolites and enzymes and the fact that many of them were putatively produced under in vitro conditions suggests that they could also be produced in the wounds of detached fruit and activate plant defense.

Another interesting point in this study was that the difference in bacterial establishment ability between the two fruit was evident, since 36 h of incubation prior to fungal inoculation were needed for the exhibition of disease suppression in grapes compared to 1.5 h needed in tomatoes for the same bacterial concentration (10^8^ CFU/mL). In addition, excessive growth of the bacterial strain was observed in tomatoes when applied in this concentration, leading to severely decreased disease incidence compared to the control, suggesting that space occupation is an effective mode of action. When concentration of 10^6^ CFU/mL was applied, overgrowth was not observed, but the same time of bacterial incubation in the wound was still sufficient for disease suppression. The wound colonization ability after 3 days of co-incubation with *B. cinerea* was slightly higher in cherry tomatoes compared to grape berries, although the initial inoculant contained less CFU. Different colonization ability of the same strain in different detached fruit under the same conditions of incubation was also observed in the study of [14]. Although our studies show that tomato fruit seemed a more favorable environment for quick colonization, possibly due to fruit-specific conditions, the strain efficiently colonized the wound of both cherry tomatoes and grape berries. The colonization ability of Hil4 could be attributed to the reported strong swarming, swimming, and biofilm formation abilities, and the reported production of cellulase, as well as the predicted lytic enzymes from genome mining analysis, especially pectin lyases.

Following the biological control potential of endophytic bacterial strain Hil4, its plant growth promoting abilities were tested in vitro. The plant *A. thaliana* has become a model plant to study plant-microbe interactions and is readily used for screening of PGPB and PGPBE in vitro [24,115,116]. In our study, *A. thaliana* Col-0 seedlings were treated with bacterial strain Hil4, which was inoculated at distance from the root tip and on the root tip, as well as in different compartment of petri dishes. Treated seedlings demonstrated altered phenotypes compared to the control in all types of bacterial inoculation and no pathogenicity symptoms were observed in this incubation time period. When Hil4 was inoculated at distance from the root tip, it stimulated the growth of *A. thaliana* Col-0 seedlings by increasing fresh weight, root hair number and length, lateral root number, and total root area, although primary root elongation was inhibited. The root architecture characterized by decreased primary root length, but increased lateral root number and length is reported as the clumped root phenotype and has been observed in many studies concerning PGPB [24,116,117,118]. When bacterial strain Hil4 was inoculated on the root tip, the seedlings presented higher fresh weight, but extremely short primary root, indicating the role that a high bacterial inoculant could play when so close to the very young root. Studies have shown that low concentrations of auxin cause longer roots, while high concentrations result in shorter roots [117,119], so the clumped root phenotype is a considered typical auxin-dependent phenomenon [2], although it can be caused independently of auxin production [120].

It is reported that increasing bacterial concentrations provoked decreased primary root length [117,121]. Strain Hil4 also increased fresh weight and leaf area through volatile emissions. It has been reported that microbial volatile compounds such as acetoin and indole [122,123], can enhance the growth of *A. thaliana* [124], with both possibly produced by the strain, according to in vitro assays and genome mining analysis.

While tests on the model plant *A. thaliana* are highly useful as a screening method to distinguish bacterial strains that could act as plant growth promoters, other plants should be tested as well. For this reason, crop plant *S. lycopersicum* var. Chondrokatsari Messinias was selected [61], to which the bacterial strain was applied as seed biopriming and incubated in vitro. Hil4 achieved earlier germination and increased all characteristics measured after 8 days of incubation, including fresh weight, shoot length, root length, and root area compared to the untreated control plants, while its colonization on the root was verified with dilution plating and microscopy. Colonization and adherence to the root is achieved with swarming motility and biofilm formation, characteristics both strongly present in the abilities of the strains as revealed in the in vitro tests and genome mining. Alterations of root architecture, such as the increase of total root area due to additional lateral root formation and elongation, as well as the increase of root hairs in number and length, can lead to the improvement of water and nutrient acquisition, as well as tolerance to abiotic and biotic stress [2]. It would be interesting to conduct further experiments concerning the plant growth promoting effects of the strain on tomato plants under greenhouse conditions, as well as to examine its endophytic and systemic nature.

## 5. Conclusions

The results of the present multidisciplinary study suggest that the endophytic bacterial strain *B. halotolerans* Hil4 represents a promising agent for agricultural use. This holistic approach of testing plant growth promotion and biocontrol potential is not often encountered for *B. halotolerans* strains, and it enabled us to characterize and determine the potential of the strain. Genome analysis shed light on this strain’s great genetic makeup and also revealed the presence of mojavensin BGC, a cluster absent from other plant-associated *B. halotolerans*, making this strain novel. *B. halotolerans* Hil4 showed plant growth promoting abilities regarding *A. thaliana* and *S. lycopersicum* in vitro, using different inoculation methods, as well as disease protection abilities against gray mold in cherry tomatoes and table grapes. Understanding the mode of action and performing tests on multiple fruit and against several pathogens will aid in the improvement of biocontrol efficiency by using the active form of the strain, the correct inoculum concentration, and the correct quantity, as well as using it against the intended pathogen. This strain was also capable of secreting agar diffusible antimicrobial metabolites and plant defense elicitors constitutively during solid culture, among which a novel bacterial metabolite was possibly annotated, known as azelaic acid. Our study contributes new knowledge in the field of microbial plant growth promotion and biocontrol, and will facilitate future research. Further trials for intended PGP inoculants need to be conducted in greenhouse and field conditions, which will constitute future developments of the present research.

## Figures and Tables

**Figure 1 microorganisms-09-02508-f001:**
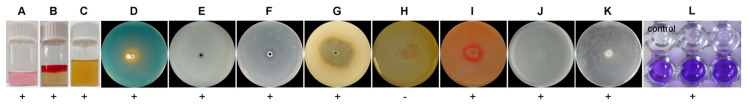
Plant growth promoting and environmental fitting traits of endophytic bacterial strain Hil4. (**A**) Indole related compounds; (**B**) acetoin production; (**C**) ammonia production; (**D**) siderophore production; (**E**) phosphate solubilization; (**F**) chitinase production; (**G**) protease production; (**H**) urease production; (**I**) cellulase production; (**J**) swarming motility; (**K**) swimming motility; (**L**) biofilm formation. A positive result is indicated by (+), where a negative result is indicated by (−). All assays were performed in triplicates in 3 independent experiments.

**Figure 2 microorganisms-09-02508-f002:**
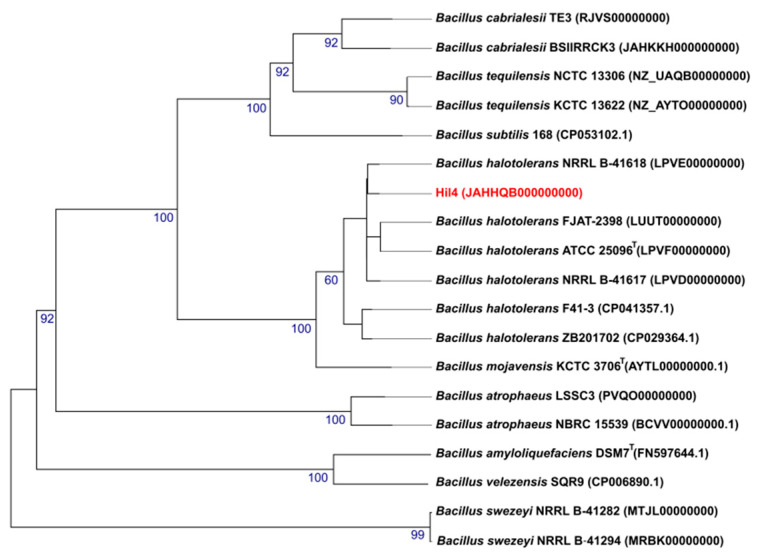
Sequence-based phylogenomic tree constructed on TYGS (https://tygs.dsmz.de/, accessed on 15 September 2021) illustrating the position of bacterial strain Hil4 (red color) relative to other closely related species. The tree was generated with FastME [36] from Genome BLAST Distance Phylogeny (GBDP) distances. The branch lengths are scaled in terms of GBDP distance formula d5. The numbers above the branches are GBDP pseudo-bootstrap support values > 60% from 100 replications, with an average branch support of 70.6%. The tree was rooted at the midpoint [51]. Type strains are indicated by ^T^.

**Figure 3 microorganisms-09-02508-f003:**
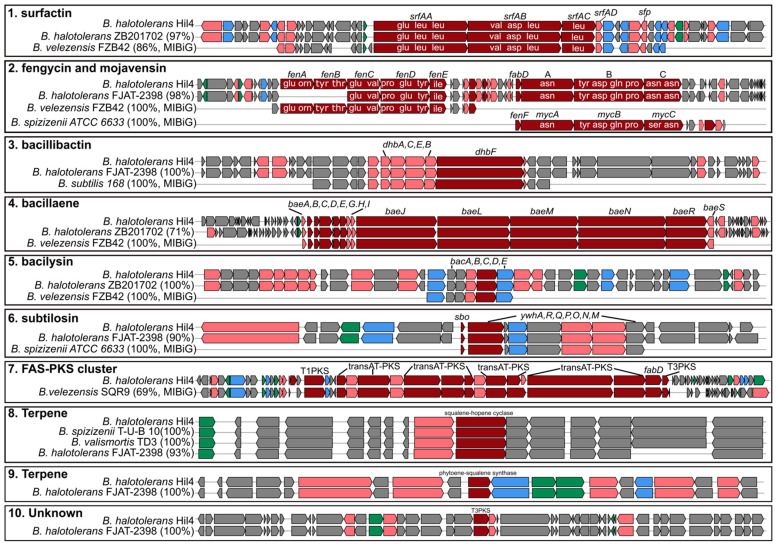
Results of the *B. halotolerans* Hil4 genome antiSMASH analysis showing genomic regions that contain biosynthetic gene clusters (BGC) of secondary metabolites and their flanking genes, along with one of the best hits in ClusterBlast and the closest core biosynthetic gene clusters of known BGC listed in the MIBiG database. Gene similarity percentage given by antiSMASH is reported in the parentheses. Core biosynthetic genes are represented with maroon color, additional biosynthetic genes with pink color, transport related genes with blue color, regulatory genes with green color, and other genes with gray color. Core biosynthetic genes are named and the aminoacid sequence of the cyclic lipopeptides is indicated with white color. More information on the BGCs and the metabolites are given in Table 2.

**Figure 4 microorganisms-09-02508-f004:**
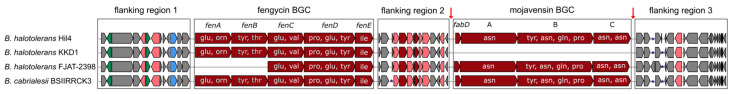
Schematic representation of fengycin/mojavensin BGCs and their microsynteny in *B. halotolerans* Hil4 and other *Bacillus* strains. The scheme was constructed based on antiSMASH ClusterBlast results, as well as BLASTn and BLASTp analysis. Core biosynthetic genes are represented with maroon color, additional biosynthetic genes with pink color, transport related genes with blue color, regulatory genes with green color, and other genes with gray color. Known core biosynthetic genes are named and predicted amino acids of the final products are noted in each synthetase gene. Each flanking region and core biosynthetic gene cluster region is enclosed in a black rectangle, while the possible site of mojavensin BGC insertion is pointed to by red arrows. Possible missing genes in flanking regions are indicated by blue asterisks.

**Figure 5 microorganisms-09-02508-f005:**
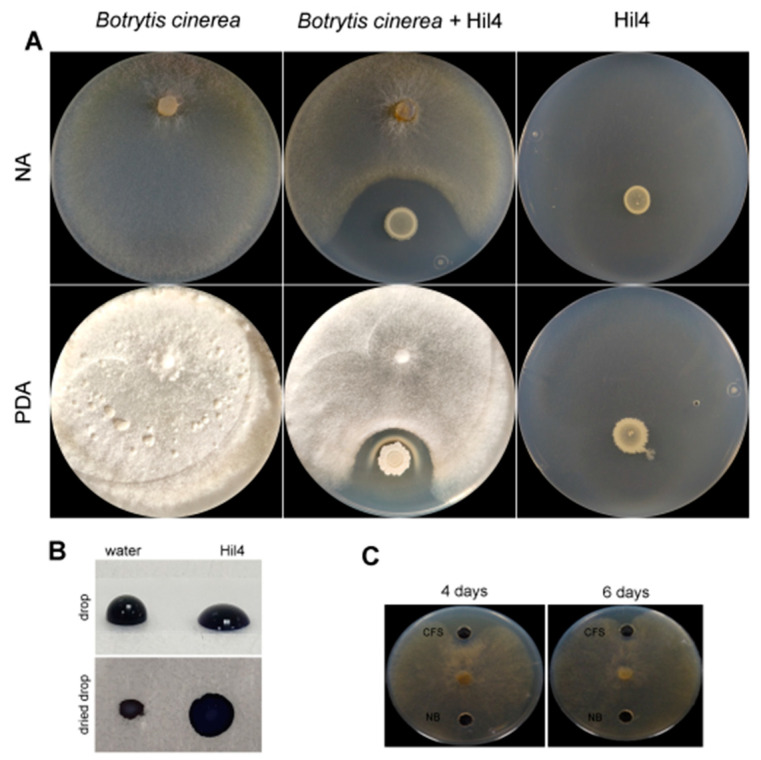
Antifungal activity of endophytic bacterial strain Hil4 against *B. cinerea* in vitro. (**A**) Dual culture on NA and PDA medium after 9 days of incubation at 25 °C, showing *B. cinerea* cultured alone, *B. cinerea* co-cultured with Hil4, and Hil4 cultured alone; (**B**) cyclic lipopeptide production testing using drop collapse assay; (**C**) activity of filtered 48-h culture supernatant (CFS) with nutrient broth (NB) as control after incubation for 4 and 6 days at 25 °C.

**Figure 6 microorganisms-09-02508-f006:**
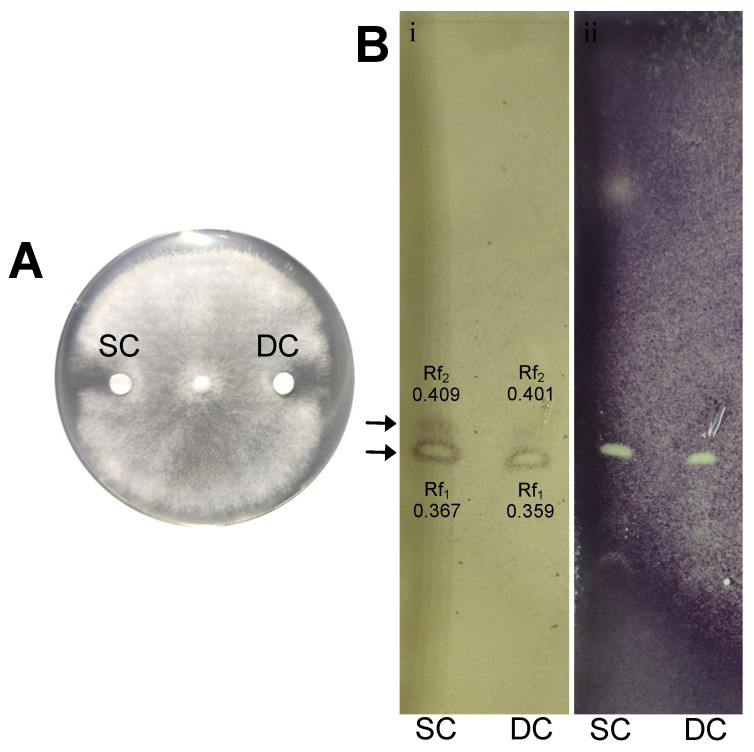
Antifungal activity and thin layer chromatography (TLC)-agar overlay bioautography analysis of ethyl acetate extracts from Hil4 single culture (SC) and dual culture with *B. cinerea* (DC) secretome on NA after 6 days of incubation at 25 °C. (**A**) Inhibitory activity of extracts from single bacterial culture (SC) and dual culture (DC) secretomes (20 μL) against *B. cinerea* after incubation at 25 °C for 4 days using the paper disc assay; (**B**) TLC and bioautography analysis of extracts. Developed chromatograms were covered with PDA containing 0.8% agar (*w*/*v*) that was inoculated with *B. cinerea* spore suspension (105 CFU/mL) and incubated for 7 (i) and 24 (ii) hours. Zones of fungal inhibition (arrows) were visualized by spraying with viability dye MTT (2.5 mg/mL). Retention factor (Rf) values are noted on the image.

**Figure 7 microorganisms-09-02508-f007:**
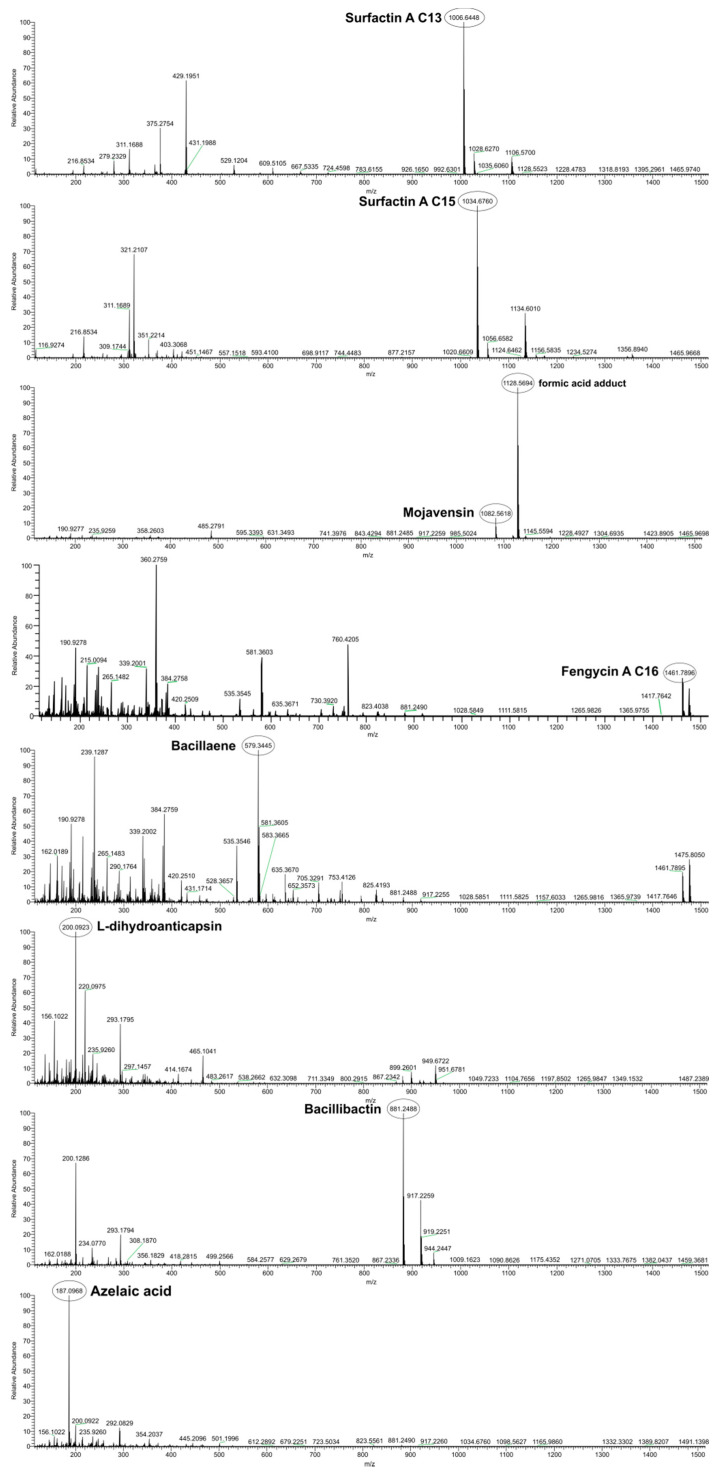
UHPLC-HRMS chromatograms of putatively annotated agar diffusible metabolites of ethyl acetate extracts originating from Hil4 single bacterial solid culture secretome. Peaks of corresponding metabolites are circled.

**Figure 8 microorganisms-09-02508-f008:**
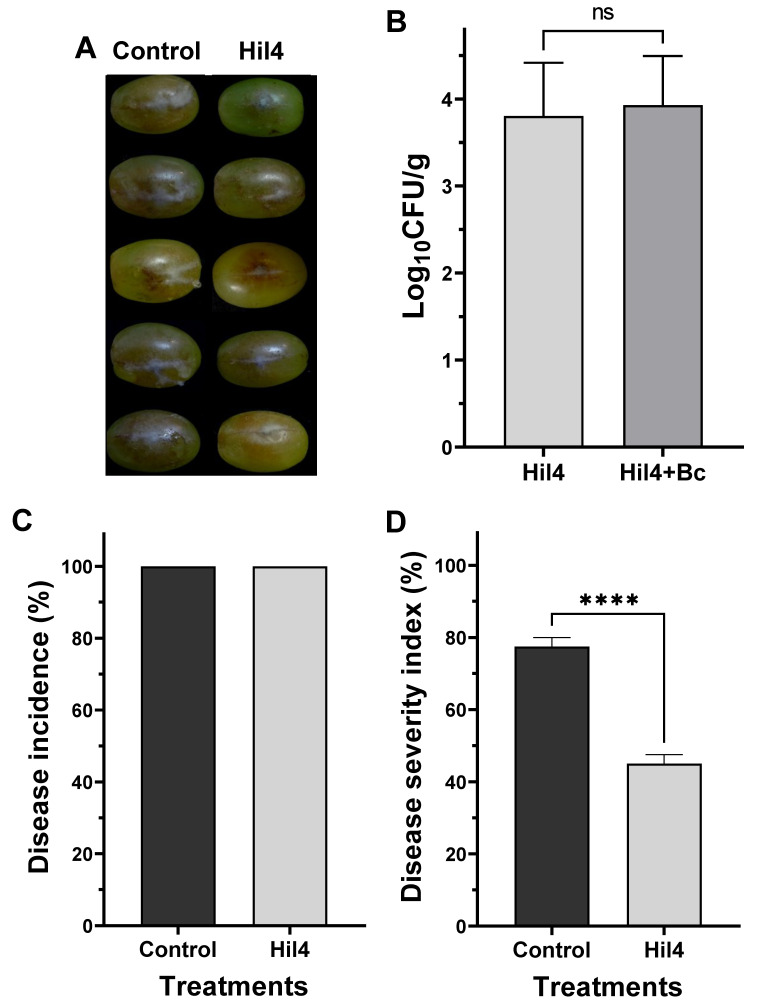
Efficacy of Hil4 culture in controlling gray mold disease of grape berries caused by *B. cinerea* in a preventive application. Artificial wounds on grape berries were inoculated with 5 μL of bacterial culture (10^8^ CFU/mL) and incubated for 36 h in the dark at 25 °C. Then, 5 μL of *B. cinerea* spores (10^5^ spores/mL) were inoculated in the same wound and fruit were further incubated for 3 days. Control treatment consisted of the pathogen inoculated alone. (**A**) Representative images of grape berries; (**B**) population dynamics of Hil4 in the wound tissue of grape berries (Log_10_CFU/g) pooled from 3 independent experiments each consisting of 6 berries (*n* = 18)d (**C**) disease incidence calculated as the percentage of infected fruit in 3 independent experiments each consisting of 10 berries (*n* = 3); (**D**) disease severity index (%), calculated in 3 independent experiments each consisting of 10 berries (*n* = 3) using a formula. Data represent mean (SD) values and asterisks indicate statistically significant differences between untreated control and treated with Hil4 after *t* test analysis (ns, non-significant; ****, *p* < 0.0001).

**Figure 9 microorganisms-09-02508-f009:**
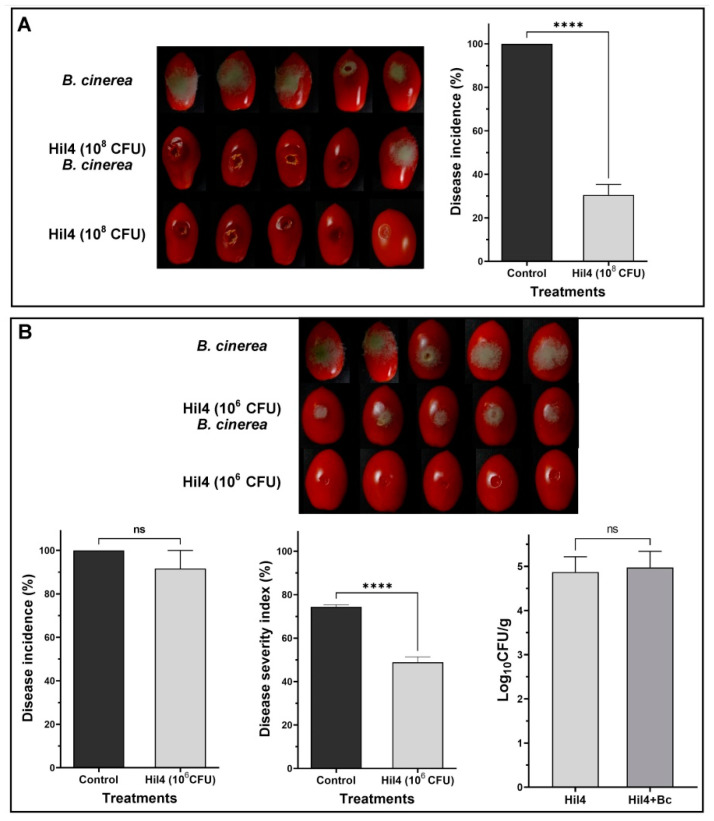
Effects of different cell concentrations from endophytic bacterial strain Hil4 on gray mold disease of cherry tomatoes in a preventive application. Artificial wounds on cherry tomato were inoculated with 10 μL of bacterial culture (10^6^ or 10^8^ CFU/mL) and incubated for 1.5 h in the dark at 25 °C. Then, 10 μL of *B. cinerea* spores (10^5^ spores/mL) were inoculated in the same wound and fruit were further incubated for 3 days. Pathogen inoculated alone served as control and the behavior of Hil4 when inoculated alone was checked as well. (**A**) Results of treatment with 108 CFU/mL of Hil4, (i) representative photos of cherry tomatoes for all treatments and (ii) disease incidence (%), calculated as the percentage of infected fruit in 3 independent experiments each consisting of 12 cherry tomatoes (*n* = 3); (**B**) results of treatment with 106 CFU/mL of Hil4, (i) representative photos of cherry tomatoes for all treatments, (ii) disease incidence (%), calculated as the percentage of infected fruit in 3 independent experiments each consisting of 12 cherry tomatoes (*n* = 3), (iii) disease severity index (%), calculated in 3 independent experiments each consisting of 12 cherry tomatoes (*n* = 3) using a formula, and (iv) population dynamics of Hil4 in the wound tissue of cherry tomatoes (Log_10_CFU/g) pooled from 3 independent experiments each consisting of 6 fruit (*n* = 18). Data represent mean (SD) values and asterisks indicate statistically significant differences after *t* test analysis (ns, non-significant; ****, *p* < 0.0001).

**Figure 10 microorganisms-09-02508-f010:**
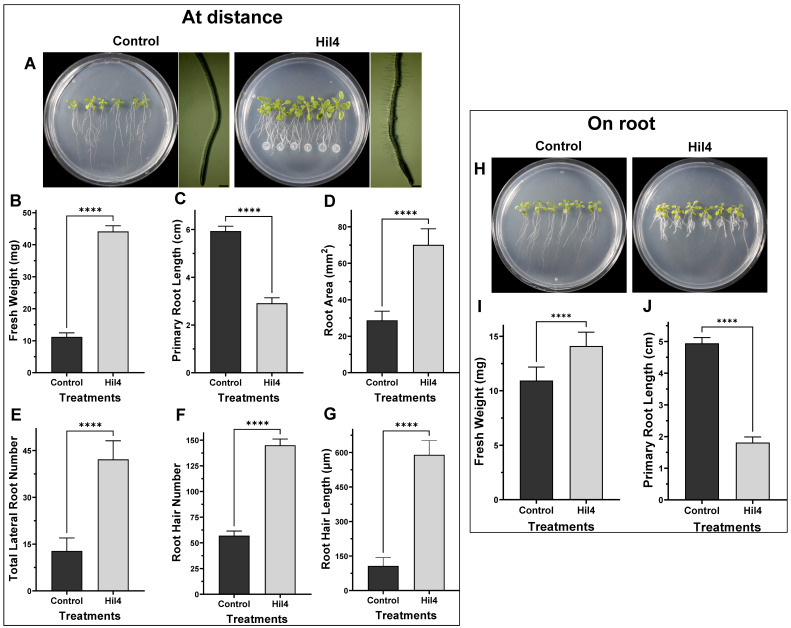
The effect of endophytic bacterial strain Hil4 on growth and root system development of *Arabidopsis thaliana* Col-0 seedlings in vitro. Four-day-old seedlings (at distance inoculation) or 6-day-old seedlings (on root tip inoculation) were either inoculated with 5 μL of bacterial suspension or left untreated (control) and further incubated for 12 days (16-h light: 8-h dark photoperiod, 22 ± 1 °C, 50–60% relative humidity). (**A**–**G**) refer to inoculation at distance, (**H**–**J**) refer to inoculation on root tip. (**A**) Representative macroscopic images of the seedlings (left panel) and stereoscopic images of the root tip (right panel, scale bar = 50 μm) for the two treatments; (**B**) fresh weight of the seedlings (mg) (*n* = 12); (**C**) primary root length of the seedlings (cm) (*n* = 12); (**D**) root area of the seedlings (mm^2^) (*n* = 12); (**E**) total lateral root number (all orders) (*n* = 12); (**F**) root hair number located 50 mm above the root tip (*n* = 12); (**G**) root hair length (μm) of the 20 longest root hair from 6 plants (*n* = 120); (**H**) representative images of the seedlings; (**I**) fresh weight of the seedlings (mg) (*n* = 12); (**J**) primary root length of the seedlings (cm) (*n* = 12). Data represent the mean (SD) of seedlings from one representative experiment. Asterisks indicate statistically significant differences after *t* test analysis (ns, non-significant; ****, *p* < 0.0001).

**Figure 11 microorganisms-09-02508-f011:**
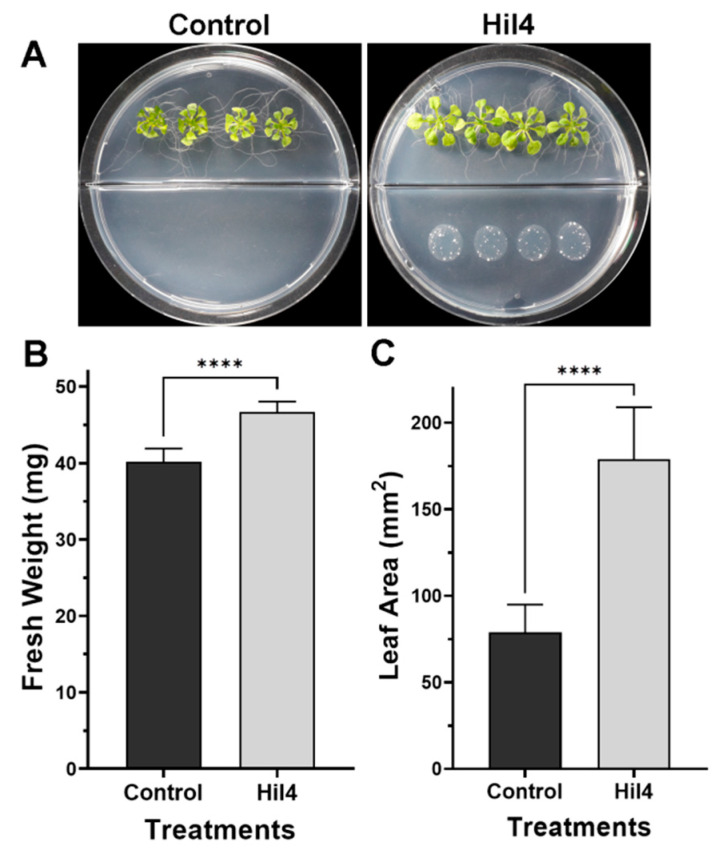
The effect of volatiles from endophytic bacterial isolate Hil4 on biomass production of *A. thaliana* Col-0 seedlings. The assay was performed using petri dishes with central partitions (I plates), where seeds were sown on one half and bacterial inoculant was placed in the other half. Measurements were made after 25 days of incubation (16-h light: 8-h dark photoperiod, 22 ± 1 °C, 50–60% relative humidity). (**A**) Representative images of seedlings for the two treatments; (**B**) fresh weight of seedlings (mg); (**C**) leaf area of seedlings (mm^2^). Data represent the mean (SD) of seedlings and asterisks indicate statistically significant difference among treatments after *t* test analysis (ns, non-significant; ****, *p* < 0.0001).

**Figure 12 microorganisms-09-02508-f012:**
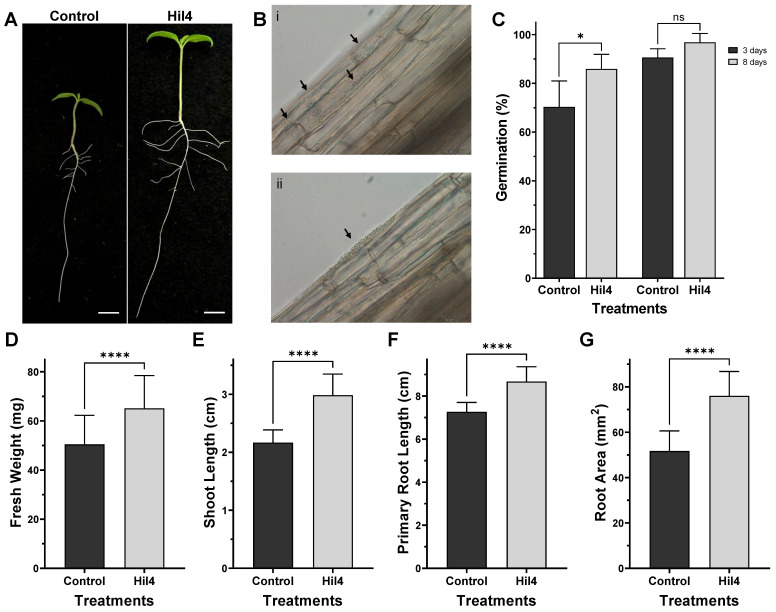
Plant growth promotion effect of endophytic bacterial strain Hil4 on *Solanum lycopersicum* var. Chondrokatsari Messinias after seed biopriming. Seeds were treated with CMC (1%) containing either Hil4 suspension (10^8^ CFU/mL) or water, sown on MS agar plates, incubated at 25 °C in the dark for 3 days to germinate, and then placed in a growth chamber (16-h light: 8-h dark photoperiod, 25 ± 1 °C, 50–60% relative humidity) for 5 additional days. (**A**) Representative images of tomato seedlings emerging from Control and Hil4 bioprimed seeds at 8 DAS (scale bar = 10 mm); (**B**) microscopic images of bioprimed seedling roots, where the arrows show bacterial cells dispersed (i) or aggregated (ii) on the root surface; (**C**) germination (%) at 3 DAS and 8 DAS from 3 replicates, each containing 16 seeds (*n* = 3); (**D**) fresh weight (mg) of tomato seedlings (*n* = 45); (**E**) shoot length of tomato seedlings (*n* = 45); (**F**) primary root length (cm) of tomato seedlings (*n* = 45); (**G**) root area (mm^2^) of tomato seedlings (*n* = 45). Data represent the mean (SD) of seedlings and asterisks indicate statistically significant difference after *t*-test analysis (ns, non-significant; *, *p* < 0.05; ****, *p* < 0,0001). The experiment was performed 3 times with similar results.

**Table 1 microorganisms-09-02508-t001:** Taxonomy of endophytic bacterial strain Hil4 based on most closely related bacterial strains belonging to closely related *Bacillus* sp. after average nucleotide identity by orthology (OrthoANI) and digital DNA-DNA hybridization (dDDH) analysis. ANI values 95–96% and dDDH values 70% are considered threshold values for species separation. Type strains are symbolized with ^T^.

Bacterial Strains	OrthoANIu (%)	dDDH (%)
*Bacillus halotolerans* ATCC 25096^T^	99.14	93.00
*Bacillus halotolerans* FJAT-2398	99.14	92.90
*Bacillus malacitensis* NRRL B-41618 *	99.26	93.50
*Bacillus axarquiensis* NRRL B-41617 *	99.17	92.90
*Bacillus halotolerans* KKD1	98.07	87.70
*Bacillus halotolerans* 36	98.07	88.20
*Bacillus halotolerans* F41-3	98.06	82.50
*Bacillus halotolerans* ZB201702	98.05	82.70
*Bacillus halotolerans* MBH1	98.05	82.10
*Bacillus mojavensis* KCTC 3706^T^	95.74	64.70
*Bacillus mojavensis* UCMB5075	95.70	64.80
*Bacillus cabrialesii* BSIIRRCK3	87.66	33.70

* They have been reclassified as B. halotolerans [49].

**Table 2 microorganisms-09-02508-t002:** Description of secondary metabolite biosynthetic gene clusters (BGCs) found in the genome of endophytic bacterial strain Hil4 (*B. halotolerans*). Genome analysis was performed using antibiotic and secondary metabolites analysis shell (antiSMASH) and the MIBiG database. PKS = polyketide synthetase, NRPS = non-ribosomal peptide synthetase.

Region	Most Similar Known Cluster	MIBIG Accession (% Gene Similarity)	Predicted Size (bp)	Synthetase Type	Metabolite
1	*srf*	BGC0000433 (86%)	65,395	NRPS	Surfactin
2	*bae*	BGC0001089 (100%)	106,300	NRPS, TransAT-PKS	Bacillaene
3	*fen*	BGC0001095 (100%)	127,469	NRPS	Fengycin
	*myc* *	BGC0001103 (100%)		NRPS, TransAT-PKS	Mycosubtilin *
4	*dhb*	BGC0000309 (100%)	47,140	NPRS	Bacillibactin
5	*bac*	BGC0001184 (100%)	41,419	Other	Bacilycin
6	*sbo-alb*	BGC0000602 (100%)	21,613	Sactipeptide	Subtilosin A
7	*art*	BGC0001520 (28%)	117,782	PKS	Unknown
8	-	-	20,807	Terpene	-
9	-	-	21,530	Terpene	-
10	-	-	40,947	T3PKS	-

* This cluster corresponds to mojavensin according to NRPS predictor 2.

**Table 3 microorganisms-09-02508-t003:** Secreted diffusible metabolites putatively annotated in ethyl acetate extract of Hil4 single bacterial solid culture secretome through UHPLC-HRMS chemical analysis.

Putatively Annotated Compounds	Molecular Formula	Calculated *m/z*	Δm (ppm)	t_R_ (min)	Adduct	References
Surfactin A C13	C_51_H_89_N_7_O_13_	1006.6435	1.29	22.68	[M-H]^−^	[53]
Surfactin A C15	C_53_H_93_N_7_O_13_	1034.6747	1.18	23.96	[M-H]^−^	[54]
Mojavensin A	C_50_H_77_N_13_O_14_	1082.5629	−1.02	14.66	[M-H]^−^	[55]
Fengycin A C16	C_72_H_110_N_12_O_20_	1461.7876	0.96	16.52	[M-H]^−^	[56]
L-dihydroanticapsin	C_9_H_15_NO_4_	200.0917	3	11.42	[M-H]^−^	[57]
Bacillibactin	C_39_H_42_N_6_O_18_	881.2471	1.91	11.89	[M-H]^−^	[58]
Bacillaene A1	C_34_H_48_N_2_O_6_	579.3428	2.93	16.49	[M-H]^−^	[59]
Azelaic acid	C_9_H_16_O_4_	187.0964	1.87	11.01	[M-H]^−^	[60]

## Data Availability

The *Bacillus halotolerans* Hil4 whole genome project is available in the NCBI database under the accession numbers JAHHQB000000000.1 (GenBank), SAMN19461816 (BioSample), PRJNA733897 (BioProject).

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
