# Peer review of "Genomic and Metabolomic Insights into Secondary Metabolites of the Novel Bacillus halotolerans Hil4, an Endophyte with Promising Antagonistic Activity against Gray Mold and Plant Growth Promoting Potential"

_microorganisms, 2021, doi:10.3390/microorganisms9122508_

Round 1

Reviewer 1 Report

In this study performed by Thomloudi et al, the authors isolated a novel Bacillus halotolerans strain Hil4 from the leaves of the medicinal plant Hypericum hircinum. Strain Hil4 exhibited antifungal activity against the fungus Botrytis cinerea. The authors performed whole genome sequencing and identified multiple gene clusters associated with plant growth promoting and anti-fungal functions.  Over all, the study was well designed, with extensive results that clearly support the conclusion. I do not see any corrections needed.

Reviewer 2 Report

The manuscript is very interesting and well addressed. The set of experiments has brought many evidences of the strain potential. Nevertheless, there are some points to improve/ precise.

1- General comments:

1.1- The introduction can be improved to gain in clarity. For instance, the first part of the introduction can be summarized and reduced, especially lines 39-56.

1.2- The fact that medicial plant was choosen for the isolation is poorly documented in the introduction. Also, while reading the paper, there is a question that shows up: why leaves were targeted for the isolation whereas you are expecting biocontrol of Botrytis on fruits/flowers or direct PGPR effect ? I think these answers will help to brush up the last part of the introduction. 

1.3- Is the strain Hil4 systemic ? I was hard to find this information.

2- Specific comments:

2.1- Please precise the commercial bleach concentration of sodium hypochloride. See lines 91, 270-271 and 314-315

2.1- How did you prepare the innoculants (bacteria and Botrytis)? With sterile water ? buffer? It was hard to locate this information.

2.2- "The homogenate was dilution-plated on..." line 308, please precise the dilution(s) used.
